# Implementing the Circular Economy by Tracing the Sustainable Impact

**DOI:** 10.3390/ijerph182111316

**Published:** 2021-10-28

**Authors:** Sebastian Lawrenz, Benjamin Leiding, Marit Elke Anke Mathiszig, Andreas Rausch, Mirco Schindler, Priyanka Sharma

**Affiliations:** Institute for Software and Systems Engineering, Clausthal University of Technology, 38678 Clausthal-Zellerfeld, Germany; benjamin.leiding@tu-clausthal.de (B.L.); marit.elke.anke.mathiszig@tu-clausthal.de (M.E.A.M.); mirco.schindler@tu-clausthal.de (M.S.); priyanka.sharma@tu-clausthal.de (P.S.)

**Keywords:** circular economy, domain-driven design, sustainability impact, service description language, digital twin, service oriented architecture, sustainable products, sustainable services

## Abstract

Sustainability is one of the most critical issues today. Thus, the unsustainable consumption of resources, such as raw materials, CO_2_ emissions, and the *Linear Economy* needs to be changed. One framework for a more sustainable economy is the *Circular Economy*. Although the concept of the Circular Economy has been around since the 1990s, yet we are still far from enabling a Circular Economy. Therefore, a turnaround to the current linear economy as well as a change in society is necessary. In this paper, we get down to the essence of the status quo in the Circular Economy, identify the main barriers, such as lack of information, unsustainable economic models, ignorance, missing incentives, and propose software-driven solutions to tackle these challenges. Our solution extends the service description language by introducing the *sustainability impact factor*. The goal is to motivate end-users towards a more sustainable behavior without making massive restrictions on their lives.

## 1. Introduction

The climate crisis is getting worse every year. We are all aware of the fact that we cannot continue ignoring this threat. However, since the climate crisis is a gradual change, it is not a topic taken as seriously by many as it should be. A total of 36,441 million tons of carbon were emitted in 2019 [1]. The Earth overshoot day 2019 fell on 29 July in 2019 [2], and humanity generated 53.6 million tons of electronic waste in 2019 [3]. We are far away from achieving a positive, *sustainable impact*.

The *Circular Economy*
*Framework*, as shown in Figure 1, claims to achieve a more sustainable approach. Unlike the linear economy, the Circular Economy focuses on keeping *things* in cycles. *Things* should be used as long as possible. The reuse of *things* has a higher priority than refurbishment and so on [4]. Recycling should be the last option for a *thing*, and options such as thermal recycling or transportation to a landfill should be avoided completely [4]. The Circular Economy framework could help us achieve a positive *sustainable impact* [4]. Even though the framework and ideas of a Circular Economy have existed since the 1990s [5], we are still far from achieving a Circular Economy as described in the literature [5].

As stated in [6], a digitized ecosystem is a beneficial enabler to achieve the idea of the Circular Economy. Our previous work defines a digital ecosystem as *an*
*open, community-driven, loosely coupled union working towards a common goal* [7]. This common goal should be the implementation of the Circular Economy. One possible way of implementing such a large ecosystem with many various actors is a *service-oriented architecture*. A service-oriented architecture develops distributed systems where the system components are stand-alone services that can deploy independently [8]. Service interfaces are usually described in a Service Description Language, such as the Web Service Description Language (WSDL). The specification of the WSDL, for example defines three aspects of a web service: what the service does, how it communicates, and where to find it [9]. *Missing in these definitions is the sustainable impact of such a service*.

### 1.1. Problem Statement

*Vision:* In an ideal Circular Economy, as shown in Figure 1, everything would stay in cycles as long as possible. This leads to more sustainability, conserves raw materials and resources, reduces the generation of electronic waste and the emission of carbon. Moreover, the sustainable impact of every action and service needs to be transparent.

*Problem:* Right now, our economy focuses on economic goals instead of ecological goals, resulting in a considerable deviation from the Circular Economy as described in the literature and reality. Furthermore, there are few targeted software engineering methods to support sustainable behavior. The sustainable impact of actions is often unknown.

*Method:* To break the barriers towards implementing the Circular Economy, we propose a sustainability impact factor that traces the environmental impact of *things*, *users, services,* and *companies*. Moreover, we present an architectural blueprint for a Circular Economic ecosystem platform that facilitates retracing and achieving the Circular Economy.

### 1.2. Objective and Contribution

*Respectively to the problem statement, the objective of this paper is to introduce a domain-driven approach to break the current barriers towards the Circular Economy*. Therefore, we analyze the status quo and the main obstacles towards the Circular Economy and propose a meta-model for a web service model including the Sustainable Impact Factor. Based on this meta-model, we derive the technical challenges for establishing our solution described as an architectural pattern and propose a blueprint for a digitized platform for a Circular Economic Ecosystem.

### 1.3. Scenario

Next, we introduce a simple scenario that focuses on several main steps during the whole lifecycle of a thing to illustrate our proposed concepts and solutions in the rest of the paper.

A factory *ToolKitCompany* buys raw materials from the mining company *SustainabilityMine* (Step 1). A *thing*, i.e., a *DrillingMachine,* is produced by an OEM *ToolKitCompany* (Step 2) and later sold to a user named *Margret* (Step 3). Margret drills ten holes in 1 h on the first day after buying the tool (Step 4). Margret has the tool for the next ten days but does not use it (Step 5). Margret’s neighbor Sam notices that she has the *DrillingMachine* and borrows the device from Margret for seven days. During that time, he drills 2.5 h which results in 25 holes (Step 6). Margret receives the tool back from Sam after a week and accidentally destroys it (Step 7). Now Margret has several options, such as throwing the device away, repairing it, or recycling it.

### 1.4. Outline

The rest of the paper is structured as follows: Section 2 analyzes the status quo of the Circular Economy and thus introduces our sustainability problem. Section 3 shows the meta-model of our sustainable impact factor and discusses the technical challenges for implementing such a solution. Section 4 discusses measurable results, and Section 5 presents an architectural blueprint for a digitized Circular Economic Ecosystem. Section 6 concludes the paper.

## 2. Circular Economy Status Quo

The goal of this section is to discuss the current problems of the implementation of the Circular Economy. As already mentioned in the introduction, there is still a substantial difference between the Circular Economy, as shown in Figure 1, and its real-world implementation. Therefore, we illustrate three perspectives: The social aspects, which consider each individual, the view of companies and politics, and the product-oriented view, which can be seen as a cross cut to the previous ones. This section focuses on the problems regarding the implementation of the Circular Economy. Note that not only should those problems be overcome, but also enablers of the Circular Economy should be supported.

### 2.1. Social Problems and Hurdles

From the perspective of an end-user/consumer, there are various reasons why they do not act optimally in the context of a Circular Economy. For some goods, especially the high-priced ones, this is evident, as they are often considered a status symbol and thus part of the lifestyle of its owner. In the segment of high-priced and luxurious consumer goods, influencing factors such as individual, psychological, cultural, and social factors play a critical role. The purpose of the product plays a subordinate role [10]. Whereby we are basically at one of the main points, the ownership. Nowadays, it is better in the sense of easier, more comfortable, and time-saving to own things instead of renting, sharing, or leasing them.

However, buying a new product instead of repairing an existing product is still more effortless, and replacement is available quickly. Without the additional time, a repairing process might require. The same is true for not using things, and it takes more effort to find someone who can use it instead of simply not using it. Even for users who are willing to make decisions in line with the Circular Economy, these decisions are not easy. These users have problems making these decisions “correctly” because the required information is often lacking [11]. On the one hand, this is due to complex and globalized production, and on the other hand, the optimum can vary locally.

### 2.2. Economic and Political Problems

Kirchherr et al. [5] address the question of which hurdles hinder or reduce the speed of change towards the Circular Economy. For this reason, the authors review appropriate literature, conduct interviews, and a survey while focusing on the European Union. Results of the mentioned work are, for example, that some of the most urgent barriers in this context are “*Low virgin material prices*” and “*High upfront investment costs*” [5]. In addition to these barriers, it is interesting to consider which hurdles exist for implementing business models that focus on sustainability and circularity. Rizos et al. [12] try to get more information about these hurdles and identify aspects supporting a Circular Economy implementation, focusing on medium-sized and small enterprises (SMEs). The article suggests the existence of indicators that a *“lack of capital”* and a *“lack of support from SMEs’ supply and demand networks”* are part of the hurdles SMEs face [12].

Hart et al. [13] identify challenges and drivers of the Circular Economy, especially regarding a specific environment. According to their article, the main barriers are “*cultural and financial*” topics, such as the fact that the sector itself is “*conservative, uncollaborative, and adversival*”. Furthermore, the authors include challenges like solving the problem of showing robust business cases in the context of a Circular Economy [13].

### 2.3. Product-Related Problems and Summary

The link between consumer and producer is marketing and advertising. In this context, advertising has a significant influence on consumer behavior and user behavior [14].

The way advertising is done changed in recent decades, driven by globalization and the rise of television and the Internet in particular. However, it is not only the technological changes in advertising but also how and what is advertised. In the 1930s and 1940s, people were still promoting throwaway tableware, later even plastic ones; today, people are moving towards grocery shopping without using outer packaging [15].

Nevertheless, sustainability is also playing an increasingly important role in advertising, as this is meanwhile a factor that influences consumers’ buying behavior [16]. You often see advertising for sales platforms, such as eBay or repair outlets, such as iFixit. Further, advertisements for sharing, leasing, or rental models can be found more and more often. However, in most advertising campaigns, the focus is still on buying the product and thus owning it [15].

On the one hand, companies build products to sell—bluntly said, a broken product leads to new products being sold. Accordingly, there is no motivation to manufacture long-lasting products and consider a design for repair and recycling. Furthermore, advertising and services for new products are present everywhere. Meanwhile, services for repair, recycling, and used products are often underrepresented. In addition to the cultural change, this also places new requirements on the design and the manufacturing of products and the associated services, such as repair or reuse. In [17], the authors identify six design strategies for sustainable products: *Product Durability, Upgradeability*, and *Adaptability* or *Design for Dis- and Reassembly*. Note that the sustainability of the production process itself has to be accounted for as well. There are indicators that *social, economic,* and *political* barriers prevent the transformation towards the Circular Economy and a high potential for improvement. Improvements are attainable once individuals make *sustainable decisions* on a regional and product-specific basis in the context of the Circular Economy lifecycle. One way to accomplish this could be a recommendation system.

## 3. Software Engineering Solution

Motivated by the problems identified in Section 2 and following the Circular Economy as shown in Figure 1, we develop the concept of the *Sustainability Impact Factor*. Thereby we overcome social, economic, and political barriers which prevent the transition towards the Circular Economy. The *Sustainability Impact Factor* is illustrated using the running case as introduced in Section 1.3.

### 3.1. Sustainability Impact Factor

The predominant linear economy does not balance economic, ecological, and social goals. Nowadays, things and services have a price (financial/economic) and a set of functionalities they offer, e.g., the *DrillingMachine* has a price, and from its functionality description, one can deduce how it works and what it can do. However, *the*
*Sustainability Impact of such a drilling machine is still unknown to consumers!*

Therefore, the *Sustainability Impact Factor* is a new sustainability dimension for *Things* and *Services.* It describes the impact a *Thing* creates on the environment throughout its *entire lifecycle*. The Sustainability Impact Factor is not just information that Things and Services should have in the proposed concept. However, it is an economic concept that motivates to decrease the environmental impacts and reduce CO_2_ emissions. There is already a solution to connect prices with CO_2_ emission. However, the pricing system in Germany is, for example, still limited to specific business sectors [18]. In the concept of the Sustainability Impact Factor, prices connect to the sustainable impact. In addition, services from several business sectors are considered as well. In contrast to other approaches, we propose a holistic approach that focuses not just on one dimension, like CO_2_. Accordingly, we define our concept as follows:

*The Sustainability Impact Factor describes a holistic and expandable concept for describing and tracing the sustainable impact of Circular Economy Entities, such as things, persons, and companies*.

Figure 2 shows the concept of the *Sustainability Impact Factor* in the context of our domain: The Circular Economy. The *meta-model* consists of a *Service*, a *Usage Record*, and a *Circular Economy Entity*. A *Circular Economy Entity* offers *service* (s) and also uses *service* (s). Specifically, a *Person* or a *Thing* is a *Circular Economy Entity*. A *Tool* and an *IT-Service* provider are specifications for a *Thing*. In contrast, a *Natural Person* or a *Legal Entity* are specifications for a *Person*. A person can also own a *Thing*.

Every *service* has a *Sustainability Impact Factor* described in two ways: *Fixed Sustainability Impact and Variable Sustainability Impact* (Figure 2). The *Fixed Sustainability Impact* is generated during various stages such as production and includes material consumption, energy consumption, or other emissions occurring. The *Variable Sustainability Impact* depends on the usage. The energy consumption, CO_2_ emission, or material loss per usage are examples of Variable Sustainability Impact. Every *Circular Economy Entity* has a *Balance of Sustainability*. This *Balance of Sustainability* is based on the Fixed Sustainability Impact and Variable Sustainability Impact either as *Debit* or *Credit* (See Figure 2). A *Person*, as well as a *Thing*, can have the *Balance of Sustainability*. The balance details are calculated in terms of debt, and credit is illustrated in the following subsection. 

### 3.2. Tracing of the Sustainability Impact Factor

The Circular Economic meta-model shows the abstract idea of the Sustainability Impact Factor as a new dimension and how things and people have a Balance of Sustainability corresponding to their impact on the environment and their associations. In this subsection, we carry forward the meta-model using our running.

The scenario introduced in Section 1.3 illustrates the steps which might occur throughout the lifecycle of a *Thing.* Nevertheless, in the scope of this paper, we focus on steps 4, 5, and 6 in more detail. Figure 3 shows these steps in particular and the state change of the *Balance of Sustainability* throughout various stages. The top row of the table in Figure 3 pertains to the steps, i.e., steps 4, 5, and 6. The left column corresponds to entities, like *Person* or *Things* and the *Sustainability Impact Factor* of the service, in case these entities provide a service. In our scenario, *Drill a hole: Service* is a service provided by the tool *DrillingMachine,* which has a Sustainability Impact Factor including the Fixed and Variable Sustainability Impact described on the left side of the table next to the *DrillingMachine*.

Calculation of the Balance of Sustainability varies based on whether it is a Variable or Fixed Sustainability Impact.

*Variable Sustainability Impact (Calculation):* Charged per usage, e.g., a DrillingMachine is produced, and a person uses it, then the Variable Impact is charged to the user per usage. In this example of the Drilling a Hole:Service, the Variable Sustainability Impact is calculated in hours, i.e., for every hour someone uses the tool, some amount of CO_2_ and energy is consumed. The Variable Sustainability Impact of the Drilling a Hole:Service per hour is multiplied by the drilling hours, and the result is charged to the user.

*Fixed Sustainability Impact (Calculation):* This is charged to the user on a proportionate basis and calculated using the integral of a function from the first to the last day that the user has the tool, called the Sustainability Impact Fraction, multiplied by the Fixed Sustainability Impact of the service for a specific resource. Formula (1) shows this calculation, and Table 1 includes the formula’s variables. The Variable and Fixed Sustainability impacts are calculated for every included resource. The considered function is always positive, monotonically decreasing and the integral from zero to infinity is one. In addition, there can be different functions for various resources and CO_2_ emissions.

The function is determined by the manufacturer of the tool and is used to share the Fixed Sustainability Impact for a specific period.
(1)$$I=\overset{t_{2}}{\underset{t_{1}}{\int }}f\left(t\right)dt\;\cdot \;S$$

Furthermore, the *Balance of Sustainability* has a *double-booking system*, i.e., every charged debt is credited somewhere.

Assumptions: The chosen example includes CO_2_, steel, and energy consumption, and not every kind of resource or environmental impact. Before step 4 occurs, a tool has some amount of *Balance of Sustainability*.

*Step 4: Margret drills ten holes on the first day*. Margert takes the tool and drills ten holes. Subsequently, she is charged with the Fixed Sustainability Impact for day one and the Variable Sustainability Impact for drilling ten holes in one hour. After step 4, the *debt* on *Margret* as Fixed Sustainability Impact for day one and the Variable Sustainability Impact for drilling ten holes is added as *credit* to the *tool*. In this particular example, the device provides the service and has an initial debt. In the double-booking system, every time someone *uses* or *has* the tool, the user is charged as debt for the impact, and the same amount is booked in the *DrillingMachines* balance as a credit.

*Step 5: Margret has the DrillingMachine for the next few days and pays a Fixed Sustainability Impact.* As in our concept, *owners* of *things* pay a fraction of Fixed Sustainability Impact when they have it, even if they do not use it. Thus, *Margret* has a debt of Fixed Sustainability Impact for ten days even if she does not drill during that time and the tool receives the associated credit.

*Step 6: Margret’s neighbor Sam notices that she has a DrillingMachine and borrows the tool from Margret for a week. During that time, he drills 2.5 h which results in 25 holes*. Margret’s Balance of Sustainability does not change when Sam borrows the tool for one week and drills for 2.5 h. Sam is charged with the Fixed Sustainability Impact of seven days and the Variable Sustainability Impact for drilling the holes. The same impact is also booked as a credit to the tool’s Balance of Sustainability.

In addition to services that generate positive debts, some lead to negative balance sheet entries. Examples include recycling services or persons collecting and separating trash correctly. After using a tool for a particular time, it reaches its end-of-life. At this point, the credits or the debts may prevail. If the device is then in debt, the owner of the tool must pay for the debt. A sustainability authority charges the owner and spends the revenues for services that support the transition towards the Circular Economy.

In summary, the concept offers the possibility that responsible parties pay for their total sustainability impact. Additionally, owners of things have an incentive to share due to not receiving a debt for the Fixed Sustainability Impact during the lending period. The proposed concept results in development towards the Circular Economy.

### 3.3. Challenges and Limitations

On the way to implementing the proposed solution, we are facing several challenges and limitations. Several technical challenges and problems must be solved to implement such a system and require tracing all relevant data and information that enable the meta-model. These challenges can be linked to typical computer science and software engineering problems and are explained in more detail in this subsection.

*Data and Information Privacy and Transparency:* For tracing the Sustainability Impact Factor, data and information are required to describe the sustainable impact of an entity or a service. Furthermore, data and information are essential in decision making in line with the Circular Economy, for example, to decide the depth of repair or disassembly decisions for a second life, remanufacturing, recycling, or so on [11,19]. Furthermore, another challenge related to the description above is the *integrity* of data and information. Such an approach tempts fraud and manipulation to better one’s position within the system. Last but not least, the overall process needs to find the right *balance* between *privacy* and *transparency*, especially concerning the general data protection regulations.

*Tracing and Identification:* To close the loop and achieve transparency as required before, tracking and identification of all parts and materials are necessary—especially when a Thing contains or consists of sub-things, such as the drilling machine containing raw materials. Therefore, methods for identifying all things are required on the one hand, and on the other hand, technology for tracking these things, especially when (sub-)things get combined to a new thing.

*Orchestration and cooperation:* Cooperation among services and entities is necessary to achieve actual value in line with the Circular Economy. Implementing the meta-model leads to a decentralized system where value creation depends on cooperation. Therefore, adaptive and self-autonomous systems are required, and orchestration mechanisms for ad-hoc and on-demand software systems integration in a decentralized and distributed context [20,21]. Moreover, it is crucial to have a straightforward interface between services and entities for the design of services and entities. Unless this interface is missing, communication and transparent data exchange are lacking. Similar challenges can be found in the area of the semantic web [22].

*Organizational Challenges*: The fundamental motivation behind the meta-model is not to penalize Circular Economy entities, such as the users or the OEMs. Instead, it creates incentives for more sustainable behavior and decision-making. A negative balance is not to be compensated by penalty payments but by recovery from efficient recycling processes.

## 4. Key Performance Indicators

*How can the success of such a proposed system be evaluated?* Every system, no matter how good the idea, needs KPIs for evaluation. This section discusses shortly how the results of our meta-model are measurable and gives a brief overview of KPIs:

*Usage Time*: One primary indicator for reducing the global warming potential of EE -devices is the lifetime (primarily, when a product is in use). Extending lifetime from all EE-products in the European Union could reduce CO_2_ emissions by 4 million tons [23].

*Repair Quotes*: Currently, manufacturing new EE-products in low-wage countries is often cheaper than repairing existing devices in high-wage countries such as Germany [7]. The cost and the fact that many products are not designed for repair or recycling result in low repair quotes [24]. Repair services will provide a positive environmental impact and extend the usage time of products.

*Collection and Recycling Rates*: Further KPIs are the collection rates of EE-products. In the European Union, the lawful collection rate of 65% (compare WEEE Directive 2012/19/E.U.) is mandatory—a rate that countries like Germany, with around 40%, are far from achieving. Furthermore, the collection rate does not describe the recovery of raw materials since EE-devices can also be hoarded or thermally recycled. Instead, the recycling rate is a better KPI, calculated by dividing the weight of the WEEE that enters the recycling/preparing for reuse facility by the weight (compare WEEE Directive 2012/19/E.U.). However, this quote is still low in the European Union, with 38.9% in 2018 [25].

Other possible KPIs, which are not discussed in this paper, are the *Earth overshoot day*, the *global e-waste monitor*, *sharing quotes*, or *production quotes* for new E.E. devices.

## 5. Evidence of Contribution to Sustainability

Based on the idea of a digitized ecosystem, we discussed the current challenges and described a meta-model, including the Sustainability Impact Factor, for supporting the Circular Economy Goals: *Reuse*, *second-life,* and *recycling* (as shown in Figure 1). Sustainable production as a pillar of Sustainable Development [26] and sustainable friendly products [27], which are also reflected in the factor, support these goals.

Furthermore, we discussed the drawbacks and challenges towards the implementation of the Circular Economy Ecosystem (Section 3.3): *Data and Information Privacy, Tracing and Identification*, *Orchestration and Cooperation*, and the *Organizational Challenges*.

This section aims to introduce an architectural blueprint as a reference model for achieving the Circular Economy goals on the one side and solving the implementation challenges towards the realization of this ecosystem on the other side.

### 5.1. Ecosystem Blueprint and Stakeholders

Figure 4 shows an overview of the blueprint for a holistic digitized *Circular Economy Ecosystem*. Before going into the technical details, we first introduce the involved stakeholder of such an ecosystem to address the Circular Economy Goals.

The Circular Economy Ecosystem consists of various (still abstract) types of the *Circular Economy Entity*. On the top of the Ecosystem in Figure 4, we show multiple stakeholders that represent the whole lifecycle of *Things* with the Circular Economy in an abstract manner—from production to recycling. (Raw Material) Suppliers*,*
OEMs*, and*
Recyclers are all *Legal Entities*. On the right side, we show the Circular Economy User Community, which is derived from the *person* who owns and uses a Product in the Circular Economy. The latter is situated on the left side. The product represents a *Thing*. Last but not least, at the bottom of the ecosystem, we list other *Legal Entities*, such as the Sustainability Authority, to verify and monitor the *Sustainable Impact Factor*, Digital Service Provider, and C.E. physical Service Provider. The latter represents companies that support the Circular Economy goals, such as repair companies.

### 5.2. Challenges and Limitations

In the last subsection, we introduced the Stakeholders of a Circular Economy Ecosystem. However, drawbacks and challenges towards the implementation of such an ecosystem still exist. For tackling these challenges, the SOLID criteria influence our system design: {Single Responsibility, Open/Closed, Liskov Substitution, Interface Segregation, and dependency inversion} principles, encapsulation, and design by contract [28].

All Circular Economy Entities are registered and linked to the Circular Economy Service Platform. Communication takes place exclusively via well-defined interfaces. Every entity registers and identifies himself via the Registry Service and transfers the service description in which the product’s use case is described (e.g., drilling holes for the product drilling machine and the specification of the *Sustainability Impact Factor*). Each service consists of a different layer. As illustrated exemplary for a Product in Figure 4, the idea behind these service layers is inspired by encapsulation (information hiding), and access is just possible via a defined interface.

Furthermore, the Digital Twin Sandbox inside the platform is designed as a sandbox [29] to guarantee privacy inside the platform, in contrast to the Registry Service, which is public. The Digital Twin Sandbox is divided into Core Services and the Orchestration Mechanism. The Core Services handle the secure data exchange among the individual Circular Economy Entities registered on the platform and the platform itself. The Digital Twin Sandbox acts like a digital twin [30], which can be understood as a replica of real-world objects or processes through virtual information of these objects existing in the real world. First applications of digital twins for lifecycle management are described, e.g., by Macchi et al. [31] or by Lim et al. [32]. Furthermore, the digital twin theory also assumes that the digital twin can exist even before the real product does so that decisions regarding materials, design, and production can be influenced.

Each *Circular Economic Entity has static and dynamic operational, state, and process data stored in its wallet* called the Digital Shadow [33]. Together with the Balance of Sustainability, this data can only be accessed via a secured privacy service. This is necessary because the wallet contains personal data (data protected by GDPR).

An orchestration mechanism can be constructed with additional information, e.g., about the construction and materials of the product and behavior and lifecycle describing models. How such an orchestration mechanism can work is described in the following subsection.

### 5.3. System Overview Behaviour

Triggered by the state change of the drilling machine into the state “broken” (Step 7 of our scenario), the orchestration mechanism is used to check which actions are now possible. So in terms of the *Circular Economy Framework (see*
Figure 1*), it can be decided whether it is more optimal to repair the drilling machine, or even to repair or reuse parts of it, or even* recycle it completely.

The Orchestration Mechanism, respecting the sandbox principle, can access the wallet information, i.e., the Sustainability Balance and the Digital Shadow of anything, person, or service offered, via the Service Registry. Based on the C.E. World Model, which describes the underlying C.E. Framework, the optimal decision for this case under the given conditions can be made.

Transparency is created because service descriptions are public, and anyone can see which ones exist and what impact they have on my personal balance sheet. On the other hand, personal data is protected by the sandbox. But the wallet information is necessary to determine the optimal state because it contains the information that allows statements about the health of the thing. Since the decision leaves the sandbox, this data remains protected. But with the help of the public service description, Margret can now use the service.

## 6. Conclusions and Future Work

The concept of a Circular Economy has been around for a long time now, but we are far away from unlocking its true potential. This paper outlines the hurdles hindering the implementation of a Circular Economy, such as social, economic, and political barriers. It introduces a software engineering-based approach to break the current barriers and introduce solutions to overcome these challenges. We also present a Sustainable Impact Factor as a new sustainability dimension for products and services, which describes the impact a product or service creates on the environment. This paper further showcases an architectural blueprint for a Circular Economic Ecosystem to enable and implement a Circular Economy. With the proposed solutions, the aim is to support the transition and implementation towards the Circular Economy. The concept of the Sustainability Impact Factor should contribute to overcome the hurdles regarding the Circular Economy implementation by demonstrating environmental impact and creating incentives for saving.

### 6.1. Costs and Benefits of our Approach

The main benefit of our approach is that we can give each stakeholder in the Circular Economy a sense of their actions and impact on the environment. The Sustainable Impact Factor sensitizes actions concerning the Circular Economy and enables the Circular Economy as described in the literature. Besides the sensitization, our circular ecosystem provides recommendations for actions, and orchestrations for the local best actions, for *things,* and the Stakeholders of such an ecosystem.

However, as already mentioned in Section 3.3, one of the main drawbacks is the loss of privacy. Our ecosystem and our approach are based on traceability as well as transparency. Therefore information and data are required. Furthermore, standards are necessary for enabling a common semantical understanding. Nevertheless, resources and energy costs are required to implement, operate, and maintain such a system and a legal, political framework.

### 6.2. Transferable Artifacts and Outlook

Finally, these benefits outweigh possible costs and risks. Our meta-model is primarily designed for the problems in EE-devices and E-waste in the Circular Economic. Still, it is easily transferrable to other domains, such as water, agriculture, and much more. The same applies to our architectural blueprint, even if things may not be as smart as EE-devices. This paper also showed a function for calculating Fixed Sustainable Impact using Formula (1). In the future, we plan to extend this and have different functions for various resources such as raw materials and CO_2_ emission. In the following steps, we will evaluate the first instance of such a proposed ecosystem, in the domain of recycling and E-waste against a broader community, to see if we can improve the KPIs mentioned in Section 4.

## Figures and Tables

**Figure 1 ijerph-18-11316-f001:**
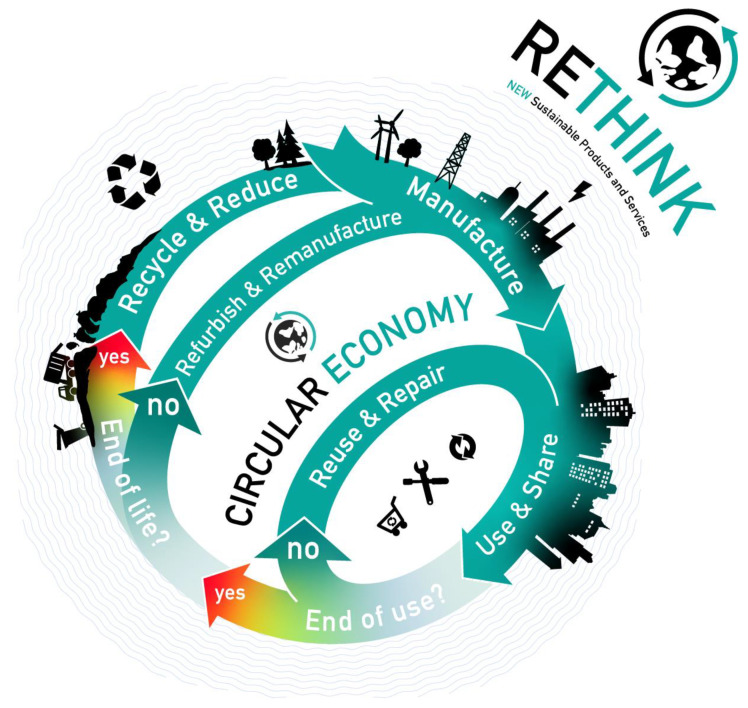
Circular Economy framework.

**Figure 2 ijerph-18-11316-f002:**
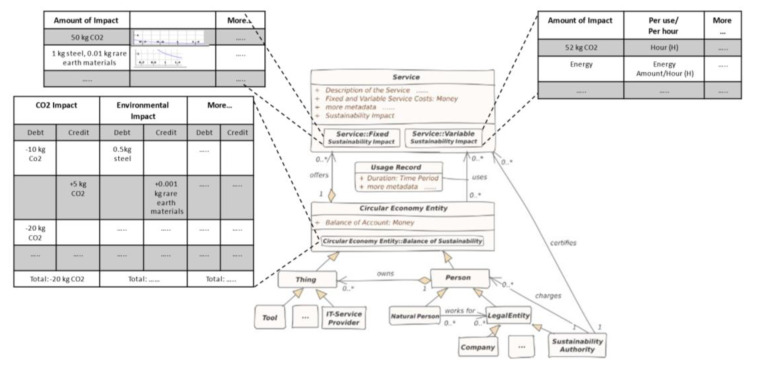
Circular Economy meta-model.

**Figure 3 ijerph-18-11316-f003:**
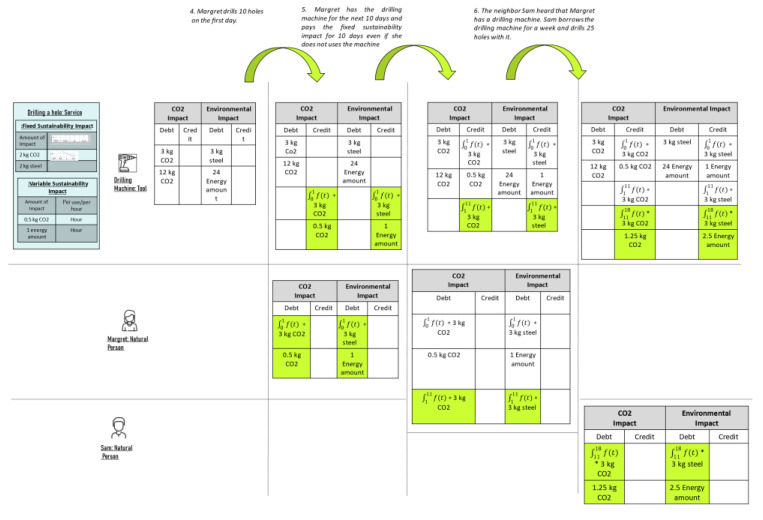
Tracing of the Sustainability Impact Factor and the meta-model example.

**Figure 4 ijerph-18-11316-f004:**
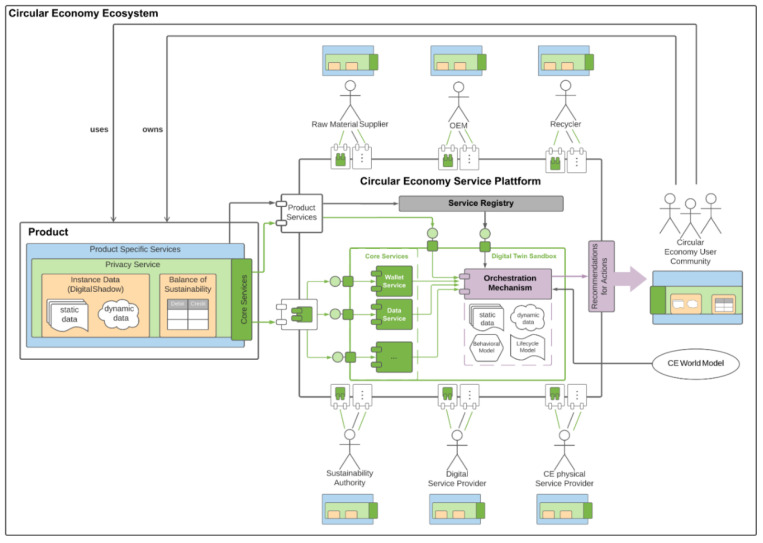
Architectural blueprint for a Circular Economy ecosystem.

**Table 1 ijerph-18-11316-t001:** Variables for Formula (1).

Variable	Description
*I*	Fixed Sustainability Impact for a specific resource that is charged to a person who had the tool for some time
*t* _1_	Start time from which a person has a tool.
*t* _2_	End time from which a person does not have a tool anymore
*f*(*t*)	Monotonically falling function for calculating the Sustainability Impact Fraction.
*S*	Fixed Sustainability Impact of a service for a specific resource

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
