# Peer review of "Implementing the Circular Economy by Tracing the Sustainable Impact"

_ijerph, 2021, doi:10.3390/ijerph182111316_

Round 1
Reviewer 1 Report
The paper presents a proposal for a software system that will support the notion of a sustainability impact factor in product design and use. I think it's a very interesting notion but I feel like the overall structure and literature review of the paper can be improved. Section 2, circular economy status quo, I found a bit lacking and I think it could be widened to include some more relevant research. For example I would suggest that the authors read the chapter 12 from the Stuart Walker is 2012 book design for sustainability as his framing of the types of obsolescence is I think very relevant to the core ideas of this paper.
In section 3 the presentation of the proposed models and the impact factor associated with it presented in a succinct way and the scenario and personas really help to get the flow through the process.
There seems to be a missing section 4. Also would urge the office to do a quick overview of contemporary strategies to foster behavioural change in relation to sustainability and product use as I feel the main finding of contemporary sustainability is that people relate much more positive codes compared to negative ones. I think reflecting on how the proposed sustainability impact assessment metric can be more effective in fostering sustainable behavioural change. I think this should also be reflected on in sections six and seven.
On the structural point of view I feel like there are a lot of jumps from section to section without making the connections clear and I would urge the authors to take a look at their content and restructure or better connect the different sections to make the paperflow more especially in the latter sections specifically 5-6-7.
Author Response
First of all, we would like to thank you for the valuable feedback!
We added some more related work, to improve the motivation, as well as Section 2. Unfortunately, we do not have access to the book by Stuart Walker (we are planning to include these in follow up work).
As suggested we also included the positive impacts. Last, but not least, we completely reworked, the story telling to avoid jumps between the sections.
Thank you again, for your valuable feedback!

Reviewer 2 Report
The article is valuable, congratulations to the authors! I have just a few minor comments.
The reference to figures in the text should be marked differently (in square brackets).
I propose to mention under the figures their source or that they have been elaborated/developed by the authors.
Line 366: In addition to “reuse, second-life, and recycling”, "reduce" is also important for sustainable development, that is, reducing the consumption of resources as much as possible.
Good luck!
Author Response
First of all, we would like to thank you for the valuable feedback!
We reoworked all the mentioned issues, and we are glad that you enjoyed the paper.

Reviewer 3 Report
The manuscript proposes an approach aimed at overcoming the barriers that prevent an effective implementation of the circular economy. I believe that this work is a valuable contribution.
However, I suggest you make some changes. Below, my comments.
Introduction. This section defines a general overview of the problem. However, it is poor in bibliographical references. Therefore, I suggest to enrich this paragraph with more references. Some examples:
line 40-41: "The reuse of things has a higher priority than refurbishment and so on."
line 43-44: "The framework of the Circular Economy could help us to achieve a positive, sustainable impact."
line 51-53: "A service-oriented architecture develops distributed systems where the system components are stand-alone services that can deploy independently."
Author Response
We want to thank you for your helpful comments and the review. We addressed the issues above by adding further references.

Reviewer 4 Report
Dear authors,
it was a great pleasure for me reading your paper which I appreciated very much.
However, I would like to point out some suggestions, hoping that you may find interesting:
- it would be very useful for the readers to have a framework about the factors that could facilitate, and not only hinder, the implementation of the circular economy;
- at the end of the article, I would like to suggest to highlight out some valuable solutions in order to overcome the limits clearly stated.
Overall, I truyly appreciated yor own definition of the concept of sustainability impact factor as well as the description of the KPI.
Author Response
We want to thank you for your helpful comments and the review. We added a sentence that contributes to the relevance of the enablers of the Circular Economy. Further a sentence was added that focuses on demonstrating the environmental impact and the creation of incentives in order to overcome the hurdles of the Circular Economy implementation.
